# A qualitative examination of injury prevention strategy and education in Ladies Gaelic football: Understanding the preferences of players and coaches

**John Corrigan**[ID]*, **Sinéad O'Keeffe, Enda Whyte, Siobhán O'Connor**

Centre for Injury Prevention and Performance, School of Health and Human Performance, Dublin City University, Dublin, Ireland

* john.corrigan23@mail.dcu.ie

## Abstract

The high rates of injury in Ladies Gaelic football impact the wellbeing of players and are a major financial burden for the Ladies Gaelic Football Association. Effective injury prevention programmes have been developed for Gaelic games, but these are not currently widely adopted. The aim of this study was to qualitatively investigate adult Ladies Gaelic football players and coaches' preferences for injury prevention strategies and injury prevention education using a constructivist grounded-theory approach. Semi-structured interviews were conducted with 23 currently active Ladies Gaelic football coaches and adult players. The core strategy preferences discussed by participants were the properties of an injury prevention programme, the role of stakeholders, strategy logistics and the need for guidance and support. If the uptake and long-term adoption of an injury prevention programme is to be maximised, the preferences of the players and coaches who will ultimately utilise the programme must be understood and integrated into a wider implementation strategy developed around their needs. For education preferences, the core categories identified were the focus of education, who needs education, the format, educator, roll-out strategy, and time required. Future educational strategies must incorporate the preferences of stakeholders into their design if they are to be successful in spreading injury prevention knowledge and achieving change. To give injury prevention strategies, programmes, and education the best chances at successful adoption, it is crucial that the preferences of end-users are addressed and implemented.

**Data Availability Statement:** All relevant data are within the paper and its Supporting Information files.

## Introduction

Ladies Gaelic football (LGF), governed by the Ladies Gaelic Football Association (LGFA), is the most played and watched women's sport in Ireland and is expanding in popularity across Europe [1, 2]. Participating in community team sports like LGF can lead to many health benefits, both physical and mental [3], however there is also an inherent risk of injury associated with participation [4]. Analysis of LGF injury claims over nine years has shown that injury

**Funding:** JC received Funding for this research from Sport Ireland, Healthy Ireland, and the Irish Research Council in collaboration with the Ladies Gaelic Football Association under grant number EPSPG/2022/395. The funders had no role in study design, data collection and analysis, decision to publish, or preparation of the manuscript. Sport Ireland: https://www.sportireland.ie/ Healthy Ireland: https://www.gov.ie/en/campaigns/healthy-ireland/# Irish Research Council: https://research.ie/ Ladies Gaelic Football Association: https://ladiesgaelic.ie/.

**Competing interests:** The authors have declared that no competing interests exist.

rates are high, at 12.98 injuries per 1000 players [5]. Injury negatively affects player health, wellbeing, and quality of life [6] and is a prominent reason given for dropping out of Gaelic games [7]. Injuries also present a substantial financial cost to players, clubs, and sport's governing bodies [5], with the LGFA Injury Fund amounting to 21% of the organisation's total expenditure in 2021 [8].

Injury prevention programmes (IPPs) specifically designed for Gaelic games players do exist, including the GAA15 [9] and Activate GAA [10] warm-up programmes. Previous research indicates that these IPPs are effective in improving neuromuscular outcomes [11, 12] and reducing injury incidence [13]. However, despite their efficacy in the literature, the actual uptake of IPPs in community sports appears to be low [14–16]. While no published research has examined the extent of IPP usage in LGF, their adoption in Camogie, a similar women's Gaelic sport, was low, with only 1 in 3 coaches report utilising an IPP within their teams [15]. This lack of usage could be explained by coaches' lack of perceived ability when it comes to conducting IPPs [15] and enhancing coaches' perceived ability through education should be a critical component of a future injury prevention strategy (IPS). An IPS for LGF players needs to be developed that aims to achieve extensive uptake and long-term adoption of IPPs.

It is well-established within the field of sports injury prevention (IP) that interventions will not be effective unless they are broadly adopted, implemented correctly, and maintained long-term [17]. Furthermore, it is understood that for an IPS to have the best chance of success, an in-depth understanding of the end-users' (e.g., coach and players) preferences and their specific implementation context is required [18, 19]. No previous research has considered the preferences of players or coaches towards IPPs in Gaelic games. In Camogie, stakeholders' preferences around the delivery of IP education have been surveyed, revealing coaches were most interested in attending practical IP courses and receiving IP information online or by post [20]. Adolescent female soccer, volleyball, and field hockey players believed IPPs should include some form of stretching, strengthening and cardiovascular activity [21]. While youth handball players requested an IPP that was sport-specific, could be completed in pairs, and took roughly fifteen minutes to complete [22]. IP preferences in Gaelic games have not been explored previously using qualitative methods. Utilising a qualitative approach when investigating players and coaches' preferences around IP allows for deeper insights into their viewpoints and more comprehensive explanations of the factors that personally affect them [23, 24]. Qualitative research gives stakeholders the opportunity to explain their preferences and their circumstances in their own words, which can lead to superior practical outcomes in the implementation of IPPs [23, 24]. Thus, the aim of this study was to qualitatively investigate adult Ladies Gaelic football players and coaches' preferences for IPS and IP education.

## Materials and methods

### Design

This study employed a constructivist grounded theory (CGT) approach, where new theory is generated from emerging data rather than pre-existing theoretical frameworks [25]. CGT advocates that individual's actively create the realities they participate in [25] and that meaning and experiences presented in research are co-constructed by the participants and researchers [26, 27]. CGT recognises researchers and participants all have multiple different viewpoints, values, realities, backgrounds, and relationships that may shape research findings [28]. This approach advances the literature by offering new explanations and concepts [29] and is fitting for this study where a gender- and sport-specific theory that specifically tackles IPS and IP education preferences in LGF is not yet established.

## Participants

Semi-structured interviews were completed with 23 currently active coaches (N = 13) and adult LGF players (N = 10). All participants were recruited via word of mouth and through social media, and informed written consent was obtained prior to commencing data collection. Theoretical sampling led participant recruitment in this study and persisted until the point of data saturation was met, which refers to the point in time when no new themes are being discovered from the data [30].

## Procedures

This project received ethical approval (#2022/031) from the Dublin City University Research Ethics Committee. A pilot study involving 5 semi-structured interviews with 5 Gaelic footballers (aged 29.0±7.7 years) was completed to increase the interviewing experience and develop the coding proficiency and analytical expertise of the primary author. The interview guide and coding procedure were deemed to elicit satisfactory responses and coding based on this pilot study. The pilot study results are not presented here with the main analysis. The authors developed the semi-structured interview guide by utilising their clinical and research experience within sports IP and following Kallio and colleagues' framework for interview guide development [31]. The guide featured several open-ended questions and additional supporting questions to gain a greater understanding of certain context-specific issues (S1 File). The primary author conducted all of the semi-structured interviews. The interviews were 48.4±12.2 minutes on average. Interviews were audio-recorded through Zoom (Zoom Video Communications, Inc., San Jose, CA) and an automatic transcript was also created. The recording and transcript were examined and amended where necessary to ensure all responses were transcribed verbatim.

## Data analysis and trustworthiness

NVivo12 (QSR International, Melbourne, Australia) was used to complete the reflexive thematic analysis [32]. Reflexive thematic analysis contains six stages of analysis (a) familiarisation, (b) code generation, (c) sub-theme/theme identification, (d) theme review, (e) theme and category finalisation, and (f) reporting findings [32, 33]. Once transcription was complete, each of the transcripts were read numerous times by the primary author so as to become familiar with the language used by participants. Initial codes were generated by breaking up and labelling data within transcripts. These were subsequently reviewed and associated sub-themes were merged into themes. Themes were narrowed/expanded versions of existing codes or new groupings that included many different codes. The relationships between themes were reviewed and analysed next, and the themes and core categories were finalised. A 'critical friend approach' was followed to maximise overall methodological rigor. The primary author (JC) held regular discussions with an evaluator independent of the interview sessions (SOK), these interactions allowed for extensive comparison and debate around their subjective interpretations of the data. This method promotes reflexivity by examining an individuals' own construction of knowledge [34], effectively improves overall transparency and trustworthiness while also ensuring the methodological soundness and practical relevance of the research [35].

Investigator triangulation, persistent observation, the use of raw quotes to convey findings [36, 37] and a negative case analysis [37, 38] were all performed to establish credibility. Detailed descriptions of the research process and the participants themselves were included to assist with transferability, as this enables the reader to assess whether findings are relevant to them and their setting [37]. An audit trail was maintained that featured all notes and decisions made during the research process to guarantee dependability and confirmability of the

research [36, 37]. The Standards for Reporting Qualitative Research Checklist [39] (S1 Table) was used in this investigation.

## Results

Thirteen coaches (8 male, 5 female) were interviewed with an average age of 46.1±5.6 (38–56) years and 14.5±6.5 (5–28) years of experience coaching in LGF. Coaches reported the highest level of LGF that they are currently coaching, with nine at club level and four at inter-county level. Sixty-nine percent (N = 9) of the coaches interviewed were coaching more than one LGF team. Ten active adult LGF players were interviewed with an average age of 26.1±4.4 (22–37) years, and with 18.6±5.0 (12–29) years of experience playing LGF. Players reported their current level of play, with two players competing at both senior inter-county and club level and eight players competing only at club level.

### IPS preferences

The core categories of strategy preferences discussed by participants were the properties of an IPP, the role of stakeholders, the logistics of an IPS and the need for guidance and support. These preferences were similar for coaches and players (Tables 1 and 2).

**IPP.** Players and coaches discussed the characteristics of potential IPPs at length and highlighted that if a programme intends *"to be long-term it must have variety and progressions year-on-year. . . the programme has to have development, and fun, it has to be enjoyable" (C2)*. They felt for an IPP to be successful it needs to target the prevention of the most common and worst injuries and be sport-specific:

> *"mimic the game as much as you can, expose us to situations and positions where we might be at risk of injury, but we're getting prepared and used to those situations and positions. . . we're rehearsing them, training them, and incorporating them into an intervention programme to be able to deal with them. The strategy has to replicate the game itself" (P3).*

Participants believe it is necessary the programme provides several options, they feel *"a warmup and cooldown need to be a priority in every session" (P5)* and that the season should *"start with a good injury prevention programme preseason, implement that, and continue it through the season" (P3)*. Specific recommendations on what they would like to see in the IPP were also made: *"touch on all elements of the sport; speed, agility, aerobic fitness, strength, mobility. They will enhance your performance, enhance your recovery, and prevent injuries" (P10)*, but it was noted that *"this has to fit nicely, so have ball work incorporated into your S&C and injury prevention, you're more likely to get buy-in then from coaches and players" (C6)*. The majority of participants called for individualised IPPs, some wanted just the level of play to be taken into account: *"this depends on the age group, it has to be all age and level specific"(C10)*, while most wanted an IPP which considers that:

> *"every player is different, what works for me might not work for you. . . it has to be tailored injury prevention, or adapt to the individual over time, you can't treat everyone the same on the team long-term" (C7).*

Conversely, some felt *"it isn't feasible to make individual programmes for everyone" (P6)* and instead called for standardised programmes: *"the LGFA are missing a set programme like the FIFA11+ or Football Australia's Perform+, that kind of programme is accessible and very easy to*

**Table 1. Core categories, themes, and sub-themes surrounding coaches' injury prevention strategy preferences (no. of coaches, no. of references).**

| Core Categories | Themes | Sub-themes | |
|---|---|---|---|
| **IPP** (13, 295 times) | Characteristics (13, 178 times) | Specific exercises included (11, 42 times) | Flexibility & mobility exercises (8, 12 times) |
| | | | Fitness and conditioning work (6, 11 times) |
| | | | Strengthening exercises (6, 10 times) |
| | | | Plyometrics & agility exercises (2, 6 times) |
| | | | Fundamental movement patterns (2, 3 times) |
| | | Includes goal setting & progression (9, 23 times) | |
| | | Fun & variety (10, 22 times) | |
| | | Accessible, adaptable, user-friendly (9, 22 times) | |
| | | Includes a warm-up protocol (8, 18 times) | |
| | | Includes a preseason or offseason intervention (7, 17 times) | |
| | | Includes a cooldown protocol (4, 11 times) | |
| | | Sport-specific, games-based, includes the ball (5, 9 times) | |
| | | Targets worst & most common injuries (6, 8 times) | |
| | | Includes several options (5, 6 times) | |
| | Type (13, 59 times) | Individualised IPPs (12, 46 times) | Specific to an individual's needs (10, 37 times) |
| | | | Specific to age group/level (6, 9 times) |
| | | Standardised IPPs (7, 13 times) | |
| | Format (12, 31 times) | Team based with individual elements (11, 24 times) | |
| | | IP completed individually (5, 7 times) | |
| | Equipment Use (10, 17 times) | Anti-/minimal equipment use (8, 11 times) | |
| | | Pro-equipment use (5, 6 times) | |
| | Training vs game day use (6, 9 times) | IPP differs between training and game day (4, 5 times) | |
| | | IPP consistent between training and game day (4, 4 times) | |
| **Role of stakeholders** (13, 97 times) | Coaches (10, 29 times) | Implementing an IPP (7, 12 times) | |
| | | Promoting IP (6, 9 times) | |
| | | Communication (5, 8 times) | |
| | Players (11, 28 times) | Taking some responsibility for their own IP (9, 21 times) | |
| | | Promoting IP (3, 4 times) | |
| | | Providing feedback on IP (2, 3 times) | |
| | LGFA (9, 17 times) | Supporting IP (6, 9 times) | |
| | | Releasing IP strategy (4, 6 times) | |
| | | Pushing IP education (5, 5 times) | |
| | Clubs (9, 16 times) | Supporting IP (9, 16 times) | |
| | H&F Professionals (6, 7 times) | Access to H&F professionals (6, 7 times) | |
| **Logistics** (11, 47 times) | IP integrated into sessions (9, 20 times) | IP at every session (7, 11 times) | |
| | IP time required per session (8, 13 times) | 10–15 minutes of IP (6, 8 times) | |
| | | 16–20 minutes of IP (2, 3 times) | |
| | | 21+ minutes of IP (2, 2 times) | |
| | IP begins at a young age (4, 8 times) | | |
| | IP completed consistently (3, 5 times) | | |
| **Guidance and support** (10, 38 times) | Load management & recovery (7, 21 times) | Load management & recovery advice (7, 18 times) | |
| | | Load management & recovery policy (2, 3 times) | |
| | Injury & IP (7, 10 times) | Injury & IP advice (7, 10 times) | |
| | General Health (5, 7 times) | Nutrition & hydration advice (5, 7 times) | |

Note: Categories, themes and sub-themes ordered by most frequently referenced. LGF = ladies Gaelic football, IP = injury prevention, IPP = injury prevention programme, IPS = injury prevention strategy, LGFA = Ladies Gaelic Football Association, H&F = health and fitness.

**Table 2. Core categories, themes, and sub-themes surrounding players' injury prevention strategy preferences (no. of players, no. of references).**

| Core Categories | Themes | Sub-themes | |
|---|---|---|---|
| **IPP** (10, 303 times) | Characteristics (10, 185 times) | Specific exercises included (9, 49 times) | Strengthening exercises (7, 15 times) |
| | | | Flexibility & mobility exercises (4, 9 times) |
| | | | Fundamental movement patterns (4, 9 times) |
| | | | Fitness and conditioning work (6, 8 times) |
| | | | Plyometrics & agility exercises (5, 8 times) |
| | | Accessible, adaptable, user-friendly (8, 25 times) | |
| | | Fun & variety (9, 20 times) | |
| | | Includes a warm-up protocol (8, 20 times) | |
| | | Sport-specific, game-based, includes the ball (8, 16 times) | |
| | | Targets worst & most common injuries in LGF (9, 13 times) | |
| | | Includes a preseason or offseason intervention (6, 13 times) | |
| | | Includes goal setting & progression (6, 11 times) | |
| | | Includes several options (5, 9 times) | |
| | | Includes a cooldown protocol (4, 6 times) | |
| | | Evidence-based (3, 3 times) | |
| | Type (10, 37 times) | Individualised IPPs (9, 23 times) | Specific to an individual's needs (8, 15 times) |
| | | | Specific to age group/level (5, 8 times) |
| | | Standardised IPPs (7, 14 times) | |
| | Format (9, 34 times) | Team based IP with individual elements (7, 21 times) | |
| | | IP completed as a team (8, 13 times) | |
| | Equipment use (8, 27 times) | Anti-/minimal equipment use (7, 14 times) | |
| | | Pro-equipment use (5, 13 times) | |
| | Training vs game day use (9, 20 times) | IPP consistent between training and game day (8, 14 times) | |
| | | IPP differs between training and game day (4, 6 times) | |
| **Role of stakeholders** (10, 85 times) | Coaches (7, 28 times) | Implementing IP (7, 14 times) | |
| | | Promoting IP (5, 14 times) | |
| | LGFA (7, 22 times) | Releasing IP strategy (3, 7 times) | |
| | | Pushing IP education (5, 6 times) | |
| | | Supporting IP (4, 6 times) | |
| | | Promoting IP (3, 3 times) | |
| | Players (7, 16 times) | Taking some responsibility for their own IP (7, 16 times) | |
| | H&F Professionals (4, 10 times) | Access to H&F professionals (3, 5 times) | |
| | | Implementing IP (3, 5 times) | |
| | Clubs (6, 9 times) | Supporting IP (6, 9 times) | |
| **Logistics** (10, 59 times) | IP integrated into sessions (10, 23 times) | IP at every session (9, 12 times) | |
| | IP time per session (9, 16 times) | 10–15 minutes of IP (6, 8 times) | |
| | | 16–20 minutes of IP (4, 5 times) | |
| | | 21+ minutes of IP (2, 2 times) | |
| | IP begins at a young age (6, 10 times) | | |
| | IP completed consistently (5, 10 times) | | |
| **Guidance and support** (6, 28 times) | Load management & recovery (6, 17 times) | Load management & recovery advice (4, 9 times) | |
| | | Load management & recovery policy (4, 8 times) | |
| | General Health (3, 11 times) | Nutrition & hydration advice (3, 6 times) | |
| | | Sleep advice (2, 5 times) | |

Note: Categories, themes and sub-themes ordered by most frequently referenced. LGF = ladies Gaelic football, IP = injury prevention, IPP = injury prevention programme, IPS = injury prevention strategy, LGFA = Ladies Gaelic Football Association, H&F = health and fitness.

*follow" (C6)*. Primarily coaches and players requested for IPPs to be a mixture of team based and individual activities:

> *"It could be all part of one programme, there could be individual pieces, as well as team-based work. Don't have it all done independently, cover it in training too, but promote doing it independently or in small groups" (C8)*.

In addition to this, several players emphasised *"it's best if we as a team do it all together" (P5)*, whereas some coaches called for players to take on IP individually: *"for each individual athlete you are trying to get them to do their own program... coaches should try to put the onus back onto players" (C4)*. Preferences around equipment use were mixed in players and coaches, but the majority favoured *"designing this towards those who will have the very least of facilities, so no one is unable to do it" (C9)*. Similarly, the perspectives on training vs game day use of IPPs were diverse, most believed *"consistency is key when it comes to warming up and doing injury prevention, so keep things the same before training sessions and games" (P8)* but several noted that on game day *"you have less time so it will have to be shorter" (C6)* and that it might be best to *"remove the strength and do fewer sets on game day because we don't want to push too much" (P6)*.

**Role of stakeholders.** Participants felt coaches have a huge role to play and should actively implement IPPs and promote IP: *"there must be positive feedback and support from coaches, they have to place importance on injury prevention and highlight that to players" (C10)*. Coaches highlighted the importance of coach communication and that *"you need communication between the different coaches of the different clubs, age groups, codes etc. and communication with players and parents too" (C10)*. Both coaches and players discussed players taking responsibility for their own IP: *"it can't all go back to the coach... it can't be just education or resources aimed at coaches" (C8)* and *"you have to realise the importance of this and do it, players need to take responsibility" (P1)*. Participants also discussed the role of the LGFA and how the *"LGFA should publish their own injury prevention plans online" (P10)* and actively support and promote IP:

> *"the LGFA need to work within clubs and make it more widely known that injury prevention is important... and support you throughout implementing this, hold workshops and days where the focus is injury prevention, so they realize the importance of this, that's all got to be driven by the LGFA" (P3)*.

Players and coaches also stated clubs have to be supportive of future IPS: *"it's important the clubs push this... you need a coordinator of things inside clubs, somebody who makes sure that this is done and done right" (P8)*. They also felt H&F professionals should be more involved in LGF:

> *"There should be more work done by athletic therapists and physios within the LGFA, their role should be increased, they should be taking injury prevention at the start of sessions and going into detail with the players about it" (P8)*.

**Logistics.** Stakeholders called for IP activities to be carried out during sessions, with many believing IP *"should be incorporated into every session, it needs to become the norm and become part of everyone's routine" (P4)*. Most players and coaches felt *"injury prevention should take up 10–15 minutes of your session" (P6)* while some desired more time for IP: *"20 minutes or half an*

*hour of a session should be devoted to injury prevention exercises and improving fitness" (C1).* Participants also stated that with IP *"consistency is key, it has to be done consistently" (P7)* and that *"you get buy-in through developing habit and routine, so getting the girls into injury prevention at an early age is key" (P9).*

**Guidance and support.** Stakeholders believe *"recovery is often ignored" (P10)* and *"load management needs to be looked at more" (C5).* They called for greater advice and policy development: *"the LGFA should publicise more strategies and methods to enhance recovery and educate players and coaches on recovery techniques and the benefits" (P10)* and *"we need to pass a greater recognition of load management onto coaches and the club's themselves" (C10).* Participants also stated that within an IPS *"the LGFA needs something that covers sleep, recovery, hydration, and nutrition, the starting point needs to be those fundamental things" (P4).* Additionally, coaches called for greater guidance around injuries:

> *"there needs to be advice on the following 24 hours, what should they do when they go home? I think we should get some small medical advice because that knowledge can make a big difference" (C9).*

## IP education preferences

The core categories of educational preferences were similar for coaches and players (Tables 3 and 4). Participants' preferences on the focus of education, who needs education, the format, educator, roll-out and time were discussed.

**Focus of education.** Coaches and players felt *"education needs to revolve around why injury prevention is needed" (P8)* and *"the benefits overall" (C10): "it doesn't have to be just looked at as injury prevention. The programme is making you a stronger, faster, better athlete, promoting that will help" (C10).* They requested education specific to IPP's and warmups, such as *"footage of what the programme looks like" (P9)* and *"physically go through each section and break it down" (P9).* Participants called for detailed information on injury in LGF, with specific information on *"the biggest injury risks, the biggest injuries, why injuries happen and the main mechanisms" (P6)* and how *"injuries can impact your life, even outside of football" (C12).* Coaches also wanted to expand their knowledge of the female athlete, specifically they wanted to know more about the differences between male and female athletes, female anatomy, biomechanics, and physiology as well as their development and the menstrual cycle:

> *"education on the girls is key, we need more on the menstrual cycle, and everyone needs a basic level of knowledge and understanding of anatomy, how they work and their different needs at each stage of development" (C12).*

Lastly, participants thought education should provide *"exposure and knowledge about the overall well-being of the player" (C6)* and feature health and lifestyle advice across several different areas:

> *"There needs to be more guidelines for injury prevention and for the girls on what they should be eating and best practice hydration-wise. Give the girls advice on nutrition, hydration, and sleep, just how important they are. Mention taking time out and how important that is for mental well-being" (C5).*

**Who needs education.** Almost all participants highlighted that coaches and players need education on IP, with most believing *"Coach education would be most important, players and parents of juveniles need education too, but coach education is number one" (C8).* Developing education for all was recommended as well:

**Table 3. Core categories, themes, and sub-themes surrounding coaches' injury prevention education preferences (no. of coaches, no. of references).**

| Core Categories | Themes | Sub-Themes |
|---|---|---|
| **Focus of education** (13, 166 times) | IP and IP techniques (12, 90 times) | Why IP is required (10, 29 times) |
| | | IPPs and warm-ups (9, 25 times) |
| | | The benefits of IP (8, 17 times) |
| | | Best practice IP advice (6, 6 times) |
| | | Load management & recovery strategies (2, 6 times) |
| | | Functional movement patterns (3, 4 times) |
| | | The principles of S&C (3, 3 times) |
| | The female athlete (10, 31 times) | Differences between male and female athletes (9, 10 times) |
| | | Menstrual cycle (3, 8 times) |
| | | Anatomy, biomechanics & physiology (5, 7 times) |
| | | Differences across age groups & development (5, 6 times) |
| | Injury in LGF (9, 29 times) | Most common and worst injuries (6, 15 times) |
| | | Risk factors for injury (5, 6 times) |
| | | Impacts of injury (4, 5 times) |
| | | Mechanisms of injury (2, 3 times) |
| | Health & lifestyle advice (5, 15 times) | Nutrition (4, 6 times) |
| | | Hydration (3, 4 times) |
| | | Mental health (3, 3 times) |
| | | Sleep (2, 2 times) |
| **Format** (12, 56 times) | Method of delivery (12, 44 times) | Online education (6, 18 times) |
| | | In-person education (9, 13 times) |
| | | Dual delivery of education (8, 13 times) |
| | Accessible to all (4, 12 times) | |
| **Who needs education** (12, 50 times) | Coaches (11, 22 times) | |
| | Players (7, 11 times) | |
| | Clubs (6, 11 times) | |
| | LGFA officers (3, 3 times) | |
| | Parents (3, 3 times) | |
| **Time** (12, 33 times) | Length of education (9, 18 times) | Bite-size education (7, 13 times) |
| | | 30+min education programme (4, 5 times) |
| | Frequency of education (8, 15 times) | Once/twice a year (5, 11 times) |
| | | Continuous/gradual release (4, 4 times) |
| **Educator** (11, 32 times) | H&F professionals (8, 14 times) | |
| | Current/past players & coaches (6, 13 times) | |
| | LGFA officers (3, 5 times) | |
| **Roll-out** (11, 26 times) | Integrated into existing education programmes (6, 10 times) | |
| | Educate members & have them spread IP in their clubs (7, 8 times) | |
| | H&F professionals/LGFA officers spread IP (4, 8 times) | |

Note: Categories, themes and sub-themes are ordered by most frequently referenced. LGF = ladies Gaelic football,

IP = injury prevention, IPP = injury prevention programme, LGFA = Ladies Gaelic Football Association,

H&F = health and fitness, S&C = strength and conditioning.

**Table 4. Core categories, themes, and sub-themes surrounding players' injury prevention education preferences (no. of players, no. of references).**

| Core Categories | Themes | Sub-Themes |
|---|---|---|
| **Focus of education** (10, 129 times) | IP and IP techniques (10, 85 times) | Why IP is required (10, 33 times) |
| | | IPPs and warm-ups (10, 20 times) |
| | | The benefits of IP (5, 20 times) |
| | | Load management & recovery strategies (5, 6 times) |
| | | Basic or functional movements (3, 6 times) |
| | Injury in LGF (7, 27 times) | Most common and worst injuries (5, 9 times) |
| | | Impacts of injury (5, 7 times) |
| | | Mechanisms of injury (4, 6 times) |
| | | Risk factors for injury (4, 5 times) |
| | Health & lifestyle advice (2, 11 times) | Nutrition (2, 4 times) |
| | | Sleep (2, 4 times) |
| | | Hydration (2, 3 times) |
| | The female athlete (5, 6 times) | Differences between male and female athletes (3, 4 times) |
| | | Anatomy, biomechanics & physiology (2, 2 times) |
| **Who needs education** (10, 75 times) | Players (10, 36 times) | |
| | Coaches (9, 29 times) | |
| | Clubs (4, 10 times) | |
| **Format** (10, 74 times) | Method of Delivery (10, 61 times) | In-person education (9, 24 times) |
| | | Online education (6, 15 times) |
| | | Dual delivery of education (8, 12 times) |
| | | Text-based education (5, 10 times) |
| | Accessible to all (4, 9 times) | |
| | Interactive (4, 4 times) | |
| **Educator** (10, 32 times) | H&F Professionals (7, 16 times) | |
| | Current/past players & coaches (6, 11 times) | |
| | LGFA officers (4, 4 times) | |
| **Roll-out** (8, 22 times) | Educate members & have them spread IP in their clubs (7, 12 times) | |
| | H&F professionals/LGFA officers spread IP (3, 6 times) | |
| | Integrated into existing education programmes (2, 4 times) | |
| **Time** (8, 18 times) | Length of education (6, 10 times) | Bite-sized education (3, 5 times) |
| | | 30+min education programmes (3, 5 times) |
| | Frequency of education (7, 8 times) | Continuous/gradual release (4, 4 times) |
| | | Once/twice a year (4, 4 times) |

Note: Categories, themes and sub-themes are ordered by most frequently referenced. LGF = ladies Gaelic football, IP = injury prevention, IPP = injury prevention programme, LGFA = Ladies Gaelic Football Association, H&F = health and fitness.

*"Education is the most important thing, it has to be driven into clubs by the LGFA. . . they have to highlight the importance of injury prevention and educate everyone involved in the LGFA, players, management, officers, county boards, everyone has to know" (P3).*

**Format.** With regards to overall format, participants thought *"make it accessible, available, and convenient, that is a big thing for the education to succeed" (C11).* The methods used to

deliver IP education were discussed in great detail. The consensus from stakeholders was for *"an online workshop or in-person, or even both might be best, but I would say you need a hands-on workshop to get a good understanding" (P5)*. Others also called for: *"a webinar and access to resources that you can play back or read up about on the website" (C11)*, but conceded that *"for something like injury prevention, you really have to have face-to-face sessions" (C11)*.

**Educator.**   Discussions around who would be best to educate stakeholders on IP concluded that *"education should be taught by someone involved in injury prevention or performance, like strength & conditioning coaches, Athletic Therapists, or physios" (C7)*. Many also called for the involvement of players/coaches also:

> *"In the education you could have county players from past and present to act as role models. . . they've played the game, they understand what you're trying to prevent and what people might be going through so they can make it relevant to you as a player (P7).*

The potential for LGFA officers to act as educators was also considered: *"development officers definitely have a role to play in teaching it or organising it, maximising their involvement is important" (P8)*.

**Time.**   Most coaches preferred education *"in more bite sized chunks, that will be received better" (C3)*. Player preferences were mixed, with some demonstrating similar preferences as coaches and some calling for longer duration *"someone come in for an hour and talk to us about it" (P8)*. In terms of education frequency, completing education once or twice a year was discussed: *"do something on injury prevention before a season, have a talk on it, education, or something before the season starts, then halfway through have a refresher (P2)*. However, several players stated it would be best to *"just drip feed out this information over a few weeks, not all at once" (P10)*.

**Roll-out.**   In terms of rolling-out education, coaches mainly believed that it should be *"in the coach education programs the LGFA currently run. Injury prevention needs to be an integral part of those coaching courses" (C3)*. While many players and coaches proposed educating some club members and having them *"go and educate the rest and spread this, that's huge, just one person from each club or team would do" (P4)*.

## Discussion

### IPS preferences

This study aimed to examine the preferences of LGF players and coaches in relation to IPS and IP education. Players and coaches want IPPs that contain fun, variety and progression while also being adaptable, sport-specific and targeting the worst and most common injuries in LGF. Previous research in handball demonstrated similar findings, players wanted fun, handball-specific IPPs with plenty of variation [22, 40]. Coaches in female soccer felt if they could adapt IPPs and incorporate progression over time they could make IPPs more suitable for players, and for teams where limited resources exist resulting in more positive responses from players and greater adoption from coaches [4]. Stakeholders also gave their thoughts on what exercise types they would like to see which included strengthening, flexibility, plyometrics, agility, conditioning, and fundamental movement patterns. Near identical preferences were observed in AFLW research, with just the addition of balance training [14]. IPPs should be designed to meet the expectations of their end-users, meaning elements of fun, variety and sport-specificity must be incorporated and exercise preferences acknowledged.

Stakeholders desired an IPP that was individualised. Some requested age/level appropriate IPPs while others wanted programmes specific to each player's needs. For IPPs to be

successfully implemented they need to be made specific to their delivery context [14, 41], and by appropriately prescribing exercises based on individual ability and situation, it is believed that motivation can be enhanced, boredom reduced, and greater player buy-in achieved [42]. Currently in Gaelic games, IPPs are one-size-fits-all [9, 10], and while individualised IPPs may be superior these are more difficult to deliver at the community level [43]. Coaches in the similar women's community sport of Camogie have stated they lack the ability to implement current standardised programmes [15], therefore it may be unrealistic to expect coaches to conduct more complex individualised plans at present. Participants in this study and in ice hockey [44] suggested individualisation is ideal, but the absence of resources and education in clubs means it is not feasible. Workshops have been shown to be effective in improving community sport coaches' perceived ability to implement IPPs [45], thus if individualisation is to be encouraged in LGF, clear instructions on how to tailor IPPs must be incorporated into workshops provided to coaches.

The preference for a team based IPP with individual elements was shared by coaches and players in this study. Soccer coaches have previously stated that both team and individual IPP work is needed [4], yet research suggests that players often prefer IPPs completed in pairs or as a team, as they feel this increases motivation and enjoyment [22, 46]. Implementing an IPP that contains both team and individual exercise does effectively reduce injury incidence in professional soccer [47]. Most research to date recommending a mixed approach appears to be from coaches in elite soccer [47, 48], whereas research that looks at sub-elite settings [22] and the opinions of players [46] seems to indicate a preference for more team-based IP. Consistent with previous views in the literature [4, 14, 22], participants called for little equipment to be used in future IPPs explaining that this adds unnecessary complexity and could impede those who do not have access to equipment from completing the programme. Thus, in order to maximise adherence in a community sport setting, a team based IPP that includes individual elements and only minimal equipment should be considered.

Stakeholders felt implementing some form of IPS from an early age in LGF would be best. This has been demonstrated to make IP a more accepted part of routine and culture, prompting changes in behaviour and attitudes towards IP [49, 50]. Players and coaches also called for consistency over time in IP implementation, which is linked with positive behaviour changes in rugby players [51], enhanced long-term adherence in soccer [4] and is thought to increase chances of IPS success [23]. IPPs need to be implemented in a consistent manner from an early age in LGF if they are to facilitate uptake and prolonged adherence. Warm-ups, cool-downs, and preseason interventions were also desired by stakeholders. Cross-country coaches similarly advocate for consistent completion of team warm-ups and cooldowns as they feel this leads to greater overall programme buy-in as well as enhanced team cohesion, athlete satisfaction and performance [52]. Research shows the benefits of an IPS are maximised when activities are adopted in pre-season and continued throughout the playing season [3]. As lack of time is one of the most frequently named barriers to IPP implementation [4, 22, 40, 46, 51], future IPPs need to respect the time stakeholders are willing to dedicate to IP if they are to be accepted. The consensus among stakeholders in the current study, in ice hockey [44] and in handball [22] was for 10–15 minutes of IP to be integrated into sessions. Integrating IP directly into standard sessions is believed to facilitate superior adoption [22, 40, 46, 51], and regularly completing 10–15 minutes of IP per session over a prolonged period can lead to significant reductions in injury risk [22]. Due to the limited contact time between coaches and players in community sports [23], integrating IP into sessions can lead to competing time demands [53], and some coaches may wish to avoid completing IP in sessions as they believe this takes time away from other activities and negatively impacts performance [54]. IP education must inform programme deliverers how to optimally integrate IPPs into sessions to most efficiently use

their time, highlight the importance of IP, and explain that completing IPPs can also improve player performance.

Participants felt that coaches have a major role when it comes to implementing and promoting IP. Other research has emphasised that stakeholders feel coaches must be the leaders of IP in their sport [51, 55], and that it is necessary for coaches to be directly supervising, motivating, and assisting players throughout the execution of IPPs [40, 46, 51]. However, similar to previous research [4, 44, 48], it was acknowledged that players must also take responsibility for their own IP. Giving athletes greater ownership and responsibility over their IP practices can increase their motivation and self-efficacy, leading to greater adherence and ultimately changed behaviour [4, 44, 46]. Thus, future IPS success may rely on active involvement from both coaches and players. Stakeholders wanted the LGFA and clubs to support future IPS. Research in soccer shows few coaches feel supported by their club or association when it comes to IP and significant demand exists for increased instruction and feedback from them surrounding IPP execution [4]. Findings in the current study mirror those in soccer [4] and handball [40], where stakeholders called for their governing body to produce and promote IP educational material to improve the uptake of IPPs. Participants requested the LGFA develop and endorse their own comprehensive LGFA-specific IPS similar to those found in other sports [14, 20] to encourage more widespread adoption. Organisational support, collaboration, and endorsement is crucial in promoting IPP buy-in at all levels of sport [14, 40].

Participants called for increased accessibility and a broadening of H&F professionals' roles in LGF. IP research across several sports [23, 40, 44] has featured requests from stakeholders for increased contact with athletic therapists/trainers, physiotherapists, and S&C coaches. LGF stakeholders, as well as those in rugby [51] and soccer [42], regard H&F professionals as the most qualified to implement IPPs and the preferred educators in IP education. Their involvement is proposed to improve player motivation, exercise quality, and adherence resulting in greater initial adoption and long-term maintenance [51]. However, teams in community sports, such as LGF, lack frequent access to H&F professionals and thus decisions concerning injury [56] and load management [23] are frequently left to coaches and players. Access to medical staff in women's community sports needs to be increased and their roles expanded to include education and IPP implementation, but unfortunately the funding necessary to provide widespread medical support is frequently lacking [57]. To overcome this lack of access, stakeholders have requested guidance and policies around load management, recovery, injury, and general health. Until governing bodies can provide adequate funding, the most practical approach may be to give coaches and players information on IP and the appropriate responses to potential injury.

## IP education preferences

Players and coaches wanted greater knowledge on why IP is important, the benefits of IP, and details of IPPs and warmups. End-users in handball [40], field hockey [23], AFLW [46], and soccer [4] desired an increased understanding of the importance of IP and greater awareness of its benefits. Informing stakeholders of reductions in injury rate or risk are crucial, as female athletes are more willing to complete IPPs if shown it will result in fewer injuries, while demonstrating performance benefits also encourages adoption but may be less impactful [21]. In Camogie, 92% of coaches and 83% of players wanted education on IPPs [15], and 100% of handball coaches surveyed called for IPPs to be featured in coach education [40]. End-users are most concerned by the *what, why, and how* of IP and IPPs [46], if future education in LGF can adequately address these questions subsequent interventions may achieve superior uptake and compliance.

Stakeholders wanted increased awareness of injury in LGF and coaches in particular requested increased education on the female athlete. This is comparable with what has been proposed in Camogie [15] and rugby [58], where educating stakeholders on the risk factors, mechanisms, and impacts of common injuries as well as their early management is believed to be effective in highlighting the importance of IP and encouraging IPP use. Similar to LGF coaches, AFLW stakeholders called for more education on the menstrual cycle [46] and other factors specific to women's health [14]. Advancing the knowledge of coaches in these areas, allows them to gain a deep understanding of the importance of IP and an awareness of different elements that can affect implementation and player health.

In-person, online and dual delivery methods of IP education are preferred. A variety of format preferences for formal education and resources such as online infographics, videos, and website resources [4, 14, 15, 20, 22, 46], phone applications [4, 22], physical handouts/manuals [4, 14, 59, 60] and in-person workshops/seminars [4, 14, 45, 59, 60] have been previously highlighted by stakeholders. Stakeholders in this study and in previous literature [14, 20, 22, 45] felt that providing several different formats of educational material was best practice. However, in-person activities were emphasised by most participants as necessary for the success of a strategy in LGF, a view mirrored in netball [59] and Camogie [15] coaches. Workshops are effective at enhancing coaches' attitudes towards IPPs as well as their willingness and perceived ability to carry out IPPs [45], and online resources can be convenient and efficient tools that can be used to reinforce understanding [20] and promote further learning [45]. Many felt embedding IP education into existing education systems might be the most efficient method of dissemination. Implementing IP education into general coach education is believed to enhance the spread of IPPs [4], and has been recommended across Camogie [20], soccer [4, 60], rugby [58] and netball [59, 60]. Integrating education can boost long-term use of IPPs as it allows future coaches to be informed of the need to conduct IPPs [4]. Community sport organisations like the LGFA must design IP workshops and supporting online educational resources to ensure coaches have the necessary knowledge, skills, and motivation to implement IPPs with their teams. These must be officially incorporated into existing mandatory foundation coaching courses to ensure a minimum amount of IP awareness across the LGF community and enhance the likelihood of IPP uptake. Other suggested strategies involve different club members, H&F professionals or LGFA officers spreading IP and acting as popular opinion leaders (POLs) in LGF. These POLs would be individuals who are educated on IP that would advocate for IP at a local level and transfer key knowledge to stakeholders with the intention of changing community norms [61]. Previous research indicates that POLs can boost the speed at which new information spreads and new behaviours are adopted in communities [62]. The use of POLs throughout the country to further broadcast the importance of IP and encourage programme adoption should be considered.

Education for coaches and players was frequently requested by participants. While much of the previous research tends to focus on coach education [4, 20, 40, 49, 59, 60], many studies have developed or called for intervention strategies that inform both coaches and players [14, 22, 23,44–46, 55, 63]. Stakeholders also mentioned the value of education for all and felt that some level of education should be available and known by LGFA officers, club volunteers and the parents of youth players. This approach could create a more comprehensive strategy that promotes greater compliance among players and coaches and increases overall IPS effectiveness [64]. An educational strategy should focus primarily on coaches and players but IP educational material must be made available for all to give everyone the opportunity to enhance their awareness of IP and thus improve the strategy's chances at success. Participants also felt that past/present LGF coaches and players could play an invaluable role as educators and role models. Research in netball recommends high-profile coaches be involved in coach education

programmes as they can act as peer role models and encourage other coaches to deliver IP education to their players [65]. Handball players found it inspiring if well-known coaches or players were involved in the strategy, actively coached them on IP, or advocated for IPPs [22]. However, research into role models for IP indicates that if community-level coaches or players cannot identify with the superior status of the role model then they will have little real impact [66]. Choosing the correct educators and role models for IP education and resources is critical, stakeholders must trust that those educating them are experts in IP and/or must be capable of identifying with them directly if they are to be effective in promoting IPP adoption.

Participants called for education to be interactive, practical, and accessible to all. Coaches in Camogie [20] and netball [59] believed maximising the number of practical elements and utilising clear accessible language was highly beneficial. Previous research on IP education for youth sports [63] is in agreement with LGF coaches' opinions on education length and frequency and their preference for bite-sized education a few times per year. Other studies [20, 45, 59] recommend longer (1–2 hours) but more infrequent educational sessions. Defining the time needed for IP education in LGF is difficult and with such variety in participants' preferences it is unlikely future education strategies will match everyone's expectations. However, it is clear the potential for shorter sessions was discussed most frequently and that stakeholders wanted the minimum number of engagements required to gain the necessary IP knowledge and skills. Overall, making IP education as accessible as possible and designing workshops and resources in accordance with the preferences of participants will improve the initial uptake of education and enhance the likelihood of overall IPS success.

## Limitations

This study was the first to qualitatively investigate the preferences of LGF players and coaches towards injury prevention strategy and education, however it does possess some limitations. Firstly, the players interviewed were exclusively women, amateurs and competing in the community sport of LGF in Ireland and thus extrapolating these results to other populations and settings should be done with care. Secondly, by looking exclusively at the preferences of adult players and coaches in LGF, and not including juvenile players, the preferences of a large portion of the playing population are not represented, therefore future research should examine this in a juvenile population.

## Conclusions

Based on the preferences of LGF players and coaches, future IPPs in community sports such as LGF should be sports-specific, fun, accessible, progressive, and largely team-based with some individual elements. IP activities should start from a young age and be integrated into regular sessions for 10–15 minutes. Coaches should lead and promote IP, but players must also take some responsibility. The preferred focus of educational content was the importance and benefits of IP and how to effectively implement IPPs and warmups. This study suggests coach and player education are most important, and that stakeholders desire relatively short IP workshops guided by H&F professionals that are embedded within existing coach education. The development of IP knowledge and awareness among LGF stakeholders is crucial if future interventions are to achieve widespread uptake and compliance. The effectiveness of future strategies in community sports will be closely linked to their real-world feasibility and the availability of funding and support from clubs and governing bodies. To maximise the chances of long-term IPP adoption and maintenance in community sports, implementation strategies and supporting educational interventions must be constructed using the preferences of stakeholders.

## Supporting information

**S1 File. Interview guide questions for players and coaches.**
(DOCX)

**S1 Table. Standards for Reporting Qualitative Research Checklist.**
(DOCX)

**S2 Table. Coding framework.**
(DOCX)

## Acknowledgments

The authors want to sincerely thank all players and coaches who voluntarily gave up their time to contribute to the research.

## Author Contributions

**Conceptualization:** Enda Whyte, Siobhán O'Connor.

**Data curation:** John Corrigan.

**Formal analysis:** John Corrigan, Sinéad O'Keeffe.

**Funding acquisition:** John Corrigan, Enda Whyte, Siobhán O'Connor.

**Investigation:** John Corrigan.

**Methodology:** John Corrigan.

**Project administration:** John Corrigan, Siobhán O'Connor.

**Resources:** John Corrigan.

**Software:** John Corrigan.

**Supervision:** Sinéad O'Keeffe, Enda Whyte, Siobhán O'Connor.

**Validation:** Sinéad O'Keeffe, Siobhán O'Connor.

**Visualization:** John Corrigan.

**Writing – original draft:** John Corrigan.

**Writing – review & editing:** John Corrigan, Sinéad O'Keeffe, Siobhán O'Connor.

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
