## [Decision Letter · Decision Letter 0]

27 Jan 2023

PONE-D-22-32715A qualitative examination of injury prevention strategy and education in Ladies Gaelic football: Understanding the preferences of players and coachesPLOS ONE

Dear Dr. Corrigan,

Thank you for submitting your manuscript to PLOS ONE. After careful consideration, we feel that it has merit but does not fully meet PLOS ONE’s publication criteria as it currently stands. Therefore, we invite you to submit a revised version of the manuscript that addresses the points raised during the review process.

We look forward to receiving your revised manuscript.

Kind regards,

Ender Senel, PhD

Academic Editor

PLOS ONE

Journal Requirements:

Reviewers' comments:

Reviewer's Responses to Questions

**Comments to the Author**

1. Is the manuscript technically sound, and do the data support the conclusions?

Reviewer #1: Yes

Reviewer #2: Yes

2. Has the statistical analysis been performed appropriately and rigorously? 

Reviewer #1: Yes

Reviewer #2: Yes

3. Have the authors made all data underlying the findings in their manuscript fully available?

Reviewer #1: Yes

Reviewer #2: Yes

4. Is the manuscript presented in an intelligible fashion and written in standard English?

Reviewer #1: Yes

Reviewer #2: Yes

5. Review Comments to the Author

Reviewer #1: I think that the research was written as a structure of the original and journal guidelines. I think that the research it can be said that the article is well written.may contribute to the sports science area.

Reviewer #2: The manuscript showed to be well organized in all parts.

The idea behind is original and well established in other team sports and validate from other studies.

Futhermore i found the manuscript well writing.

6. PLOS authors have the option to publish the peer review history of their article (what does this mean?). If published, this will include your full peer review and any attached files.

Reviewer #1: No

Reviewer #2: No

---

## [Author Response · Author response to Decision Letter 0]

30 Jan 2023

Thank you so much to both reviewers and the academic editor for taking the time to read and review the manuscript. The comments were very helpful and we appreciate the effort made to do this. We have included responses to each comment in the attached document titled 'Response to reviewers'.

---

## [Editor Report · Decision Letter 1]

2 Feb 2023

A qualitative examination of injury prevention strategy and education in Ladies Gaelic football: Understanding the preferences of players and coaches

PONE-D-22-32715R1

Dear Dr. Corrigan,

We’re pleased to inform you that your manuscript has been judged scientifically suitable for publication and will be formally accepted for publication once it meets all outstanding technical requirements.

Kind regards,

Ender Senel, PhD

Academic Editor

PLOS ONE

---

## [Editor Report · Acceptance letter]

6 Feb 2023

PONE-D-22-32715R1 

A qualitative examination of injury prevention strategy and education in Ladies Gaelic football: Understanding the preferences of players and coaches 

Dear Dr. Corrigan:

I'm pleased to inform you that your manuscript has been deemed suitable for publication in PLOS ONE. Congratulations! Your manuscript is now with our production department. 

Kind regards, 

on behalf of

Dr. Ender Senel 

Academic Editor

PLOS ONE